# Web use remains highly regional even in the age of global platform monopolies

**Yee Man Margaret Ng** [1]* ⊚, **Harsh Taneja** [2]⊚*

1 Department of Journalism, Institute of Communications Research, University of Illinois-Urbana Champaign, Urbana, Illinois, United States of America, 2 Institute of Communications Research, University of Illinois-Urbana Champaign, Urbana, Illinois, United States of America

⊚ These authors contributed equally to this work.
* ymn@illinois.edu (YMMN); harsh.taneja@gmail.com (HT)

**Data Availability Statement:** Aggregations of country-level secondary data (10,083 files) that we produced into six matrices are publicly available as open-access data sources via The Illinois Data Bank (DOI: 10.13012/B2IDB-3150928_V1)—a public access repository for publishing research

## Abstract

The Internet, since its inception, has been imagined as a technology that enables information to overcome barriers of language and geography. As a handful of social media platforms now dominate globally, removing most barriers of distribution; this has created unprecedented opportunities for content to gain worldwide traction, regardless of its country of origin. Yet historically with few exceptions, people generally consume content that is from or for their region. Has the Internet or social media really altered this trait? Analyzing the extent of similarities between a hundred countries' web use patterns simultaneously across their most popular websites, and country specific trends from YouTube and Twitter respectively, we find that countries which share borders or where people speak the same languages have the most similar web use patterns. Global social media usage on both YouTube and Twitter is even more heterogeneous and driven to a larger extent by language and geography than global website traffic. Neither does high prevalence of English language speakers in the two countries, nor does one of them being the United States contributes substantially to web use similarity. Global web use remains highly regional. The technical affordances of the Internet alone are thus insufficient to render a cosmopolitan world.

## Introduction

BTS, a Korean seven-man boy band, has consistently delivered songs that topped the US *Billboard* charts, surpassing American sensations such as Kanye West [1]. To many, this is hardly surprising. The digital revolution spurred by the internet and the global adoption of smartphones, combined with the massive reach of social media, enables content to easily cross linguistic and national boundaries.

Despite this long-standing potential of the internet to render a cosmopolitan world, this has remained a largely technocratic vision. Driven by language and geography, people typically consume national rather than foreign content. In the last decade, however, a handful of platforms have amassed massive global reach that is orders of magnitude greater than what Hollywood has ever been able to achieve. Has this unprecedented worldwide availability of content,

data from the University of Illinois at Urbana-Champaign (https://databank.illinois.edu/). This service is free of charge and has a robust policy framework that fully describes the University's commitment to providing persistent and reliable access to research data and offer free and unlimited public access to data.

**Funding:** The authors received no specific funding for this work.

**Competing interests:** The authors have declared that no competing interests exist.

alongside the growth of English as a preferred foreign language, together with tools such as automated language translation, homogenized global cultural consumption at scale? If so, then phenomena such as BTS should be less of an exception and more of the norm. Yet, whether and to what extent the global footprint of the Internet in general and especially that of social media platforms enables content to transcend geolinguistic boundaries remains empirically underexplored.

The global circulation of media products began as a one-way flow primarily from large, developed countries, which dominated production infrastructure, to the less-developed ones. Western countries, especially the United States, exported television content and films to the rest of the world motivated by economic gains [2] as well as ambitions of cultural imperialism [3,4]. These theories explained the global success of Hollywood and American television [5].

The imperialism thesis is rooted in the World Systems Theory (WST), which posits the world as a complex of dependencies between core and peripheral nations. The WST envisions digital media as a site for continued western imperialism through the control of internet infrastructure. Studies mapping interconnections between countries based on hyperlinks, bandwidth, and other supply-side infrastructures repeatedly reveal the core-periphery structural hierarchies on the global Internet [6,7]. For example, most websites in the non-Western world link to Western-owned sites, which few Western sites reciprocate [8]. Likewise, pages in most languages link to English language sites with little reciprocity. Similarly, "global news flows" also reiterate this core-periphery structure [9]. Based on these asymmetric patterns of dependence, cultural imperialism and WST predict that internet content originating in Western countries will dominate global web usage.

Countering the imperialism thesis, studies of actual usage find that with the growth of domestic media industries, people generally prefer content that focuses on their region and in their own language. To be specific, media consumption is driven by cultural proximity [10]—when given a choice, people discount foreign products in favor of domestic offerings [11]. Unsurprisingly then, as Internet penetration has grown around the world, cultural proximity has explained much of global web usage. Analysis of actual usage instead of technical connectivity reveals that despite the Internet's centralized technical infrastructure, people use websites that are in the same language and focus on the same geography [12,13]. Relatedly, countries with high language similarity and shared borders tend to have similar patterns of web use [14]. Most of these studies document macro patterns of web use and hence simultaneously examine consumption of a large number of websites.

Platform-specific studies focused on usage within popular domains also echo evidence of cultural proximity. For example, people form Twitter ties based on geographical proximity, either with people in the same town or city or with those in other cities with high air traffic volumes to their own [15]. Google search trends rarely originate in the West, and their "global cultural diffusion" is relatively rare [16]. Furthermore, for the most part, users email other culturally similar users [17]. Different language editions of Wikipedia tend to be fairly diverse [18]. In summary, even for its global connectivity, web use is far from cosmopolitan [19].

Despite the overwhelming evidence for cultural proximity, it is reasonable to assume that media globalization has enabled more cross-cultural content consumption. The historical advantage of American exports in global media flows has resulted in some global convergence towards Western or American tastes. For instance, fueled by high production budgets and distribution muscle, Hollywood movies continue to dominate the global market for movie watching, even in many non-western countries [20,21].

The reach of a few social media platforms surpasses that of even Hollywood. Unlike the internet of the 1990s and early 2000s, a handful of global companies, such as Google and Facebook, the so-called platform giants [22], have unprecedented dominance over global attention

flows [23]. From being largely a network of open websites and pages connected by hyperlinks, the global web has come to be largely dominated by a handful of social media platforms that get consumed alongside the millions of websites [24]. While websites often tend to be national in character or focus, most social media provide a global platform for users to interact, thus potentially homogenizing global web use patterns.

Concerns about platform power have then rightly renewed fears of cultural imperialism. Users as well as content creators around the world, with few exceptions, flock to these same platforms, even if it is to find culturally proximate content [25]. This creates more opportunities for users to encounter content that they could not otherwise. For example, a Ghanaian user who visits YouTube for local content is quite likely to encounter a Korean video on a related topic that is also available on the same platform, possibly due to an algorithmic recommendation. Likewise, even though Facebook or Twitter users in Myanmar would primarily use the platform to connect with others in their own country, they could still easily find content overseas, especially if it is (captioned) in a language they understand. Yet studies find that interpersonal connections on social networking sites such as Facebook are driven by homophily based on language and geography [26,27].

Thus, even though web use is largely culturally aligned, content on social media could just as easily diffuse across linguistic and national boundaries. Korean pop music ("K-pop," especially Gangnam style), for instance, is popular not just in East Asia but also in South America [28]. Platforms such as YouTube, play a role in bridging otherwise dissimilar patterns of cultural consumption [29]. Content often gets discussed on Twitter beyond countries of origin.

English has defaulted to most of the world's primary or second language. This is also true of most internet users' and it dominates as the link language on Twitter, Wikipedia, and blogs online [30,31]. Thus, with English firmly established as a global bridging language, the growth of social media platforms might have contributed to homogenizing patterns of content consumption globally, at least as witnessed on these social media platforms. This argues in favor of the cosmopolitanism hypothesis. Yet the use of English as a bridging language does not necessarily imply convergence of trends between nations.

In summary, scholarly understanding of global web use remains fragmented depending on the site of research. Studies analyzing global website traffic suggest that people stick to culturally proximate websites. At the same time, it appears that consumption of content on social media platforms could overall be more homogeneous globally than website traffic.

Factors such as language similarity and geographic proximity could play a smaller role in driving web use within social media platforms than on overall website traffic. Further, there could be variations between platforms. For instance, language could play a smaller role in explaining the usage of Twitter, given the dominance English has on the platform, but could be more salient on YouTube. However, studies have yet to compare the relative power of cultural factors such as language similarity and geographic proximity in shaping countries' web usage across website traffic and content consumption within social media.

Accordingly, we examined the extent of and account for similarities between countries' web use patterns based on three different modes of online consumption. Specifically, pairs of nations, are employed as the level of analysis. Thus, in the aggregate, if people from any two countries visit or engage with the same content or outlet, we consider their web usage similar. First, we analyzed web traffic in over a hundred countries, based on people's consumption of each country's most popular websites, as well as its trending videos from YouTube and country specific trending topics on Twitter. As detailed in method, we utilized platforms' Application Programming Interface (API) and web scraping to collect two separate months of web usage data for each platform and computed pairwise similarities between countries' usage. Then, we examined the extent to which language similarity, geographic proximity, and relative size of

their Internet markets, explain the pairwise similarities between countries' consumption of each platform.

To compare website traffic between countries, we sampled each country's five hundred most-visited websites, which typically capture more than 99% of traffic. To investigate social media use at a content level, we included two platforms, YouTube and Twitter. YouTube is a global phenomenon: besides being the second most visited website in the world (behind only Google; YouTube is one of Google's subsidiaries [32]), its most popular channels posted a substantial amount of content in languages other than English [33]. Twitter is a public, real-time social platform where many users use the platform as a source of global news and use hashtags to discover posts around specific topics, allowing tracking of information flows [34]. Deeply embedded in the media ecology, the platform has served as an "unofficial extension" of traditional media ([35], p. 381). Both YouTube and Twitter arguably have a disproportionate influence on public discourse and popular culture due to the presence of many politicians, journalists, and celebrities.

Websites' country-level traffic data is readily available. However, unlike websites, individual social media platforms manifest as singular domains and do not usually provide content use statistics at a country level. Since YouTube does not provide country-level statistics for each video, we relied on YouTube "trending videos," regularly updated lists of popular videos by countries. Trending videos are determined based on several signals, including quantity and velocity of their views [36]. Thus, in each time period, the greater the overlap in two countries' trending videos, the greater their similarity in YouTube usage. Likewise, Twitter provides a regularly updated list of trending topics, in forms of hashtags, phrases, or names for different countries. Trending topics are determined by factors such as number and engagement (likes, retweets, and impressions) and tweet garners. Hashtags and phrases (e.g., #MondayMotivation and #MotivationMonday) are grouped if they are related to the same topic (#MondayMotivation; [34]). Thus, we considered two countries' Twitter usage similar based on the extent of overlap in their trending topics.

In sum, we utilized three independent metrics of relative popularity specific to each research site at the country level. From these, we computed similarity in usage patterns between countries at a pairwise level [37]. Our approach, elaborated in detail in the methods section, allowed us to compare how similar any two countries are in each time period using their relative popularity of websites with the highest traffic, their trending videos on YouTube, and their trending topics on Twitter. These pairwise similarities yielded a symmetrical country-by-country similarity matrix for each platform's usage, for each month, which became the focal variable for examining global web use similarity between nations.

## Method

### Data collection

To examine the extent to which online consumption is similar across countries, we web scraped or utilized APIs of three platforms—Alexa, YouTube, and Twitter—to obtain web usage data spanning more than 100 nations. Data for the main study was collected in September 2019. We replicated the analysis for another month—November 2019—to assess the robustness of the findings, reducing concerns that the results are artifacts of a specific month of data collection. In this section, we outline our data collection procedures for website traffic, YouTube, and Twitter respectively.

To ascertain the relative popularity of websites in a country, we obtained for each month the traffic rankings of the 500 most-visited websites for 124 countries via Alexa, a data provider that tracks web traffic through browser installations from millions of users around the world

[33]. Alexa compiles lists of the most-visited websites by country, and globally, based on monthly average web traffic. It records a site's highest-level domain (e.g., domain.com), aggregating subpages (e.g., domain.com/subpage.html) or subdomains (e.g., subdomain.domain.com). Alexa data has been used in recent communication research [6,14] and web science studies and found to have a high correlation score with ComScore's 100 most popular sites in the United States.

Querying the YouTube API (mostpopular/regioncode), we collected the first 200 trending videos for 98 countries for which data were available, four times per day. Park et al. [35] used a similar method, but they retrieved only the top 50 videos per country and did so once a day. Via Twitter API (trends/place), we collected trending topics every hour at the level of countries, for 61 available countries. S1 Table details the scale of the data collection.

While Alexa data were presented as a ranked list for each country, trending videos and trending topics from Twitter and YouTube were not. To construct a ranking list for those two platforms based on each country, we calculated the country specific ranking based on their frequency. In other words, the more often a video/topic appears in that particular country during the monthly collection period, the higher its rank. For example, since we collected YouTube data four times a day, a video could appear at most 120 times in that months' dataset. A video appearing 120 times in a month would have the highest rank, a video appearing only once would have the lowest rank. Many videos' rankings could be tied. For trending topics on Twitter, we followed the same procedure, ranking topics based on monthly frequency.

## Web use similarity

Our first step after data collection was to compute the similarity in web use between countries from their respective ranked lists of popular websites, trending YouTube videos, and trending topics on Twitter. An appropriate algorithm would look at both how many items overlap and their relative ranks. We used the Rank-Biased Overlap (RBO) algorithm developed by Webber et al. [38], which also factors the ranking position of an item in each list instead of measuring similarity solely based on presence or absence of the same items in two lists. Items overlap at each rank is weighted by a geometric sequence, providing both top-weightedness and convergence. We used the following formula to compute web use similarity between countries,

$$\mathrm{RBO} = (1 - p) \sum_{k=1}^{\infty} p^{k-1} \frac{|A_{1:k} \cap B_{1:k}|}{k}$$

$A_{1:k}$ and $B_{1:k}$ denote the top $k$ visited websites in Country $A$ and Country $B$ respectively. $p$ is a weighting parameter, which can take a value between 0 and 1.

RBO also accounts for tied ranks. Assuming t distinct items are tied for ranks $k$ to $k + (t-1)$, they all be given the same rank $k$. RBO accounts for ties by dividing twice the overlap at depth $k$ by the number of items which occur at depth $k$, rather than the depth itself. We used the following formula to calculate YouTube and Twitter similarity between countries, taking tied rank into consideration,

$$\mathrm{RBO_{tiedranks}} = (1 - p) \sum_{k=1}^{\infty} p^{k-1} \frac{2 * |A_{1:k} \cap B_{1:k}|}{|A_{1:k} + B_{1:k}|}$$

Final RBO scores range from 0 (disjoint) to 1 (identical).

Having obtained the pairwise RBO scores for all pairs of countries, we created a symmetric matrix of nations as the rows and columns where each cell would capture pairwise similarity between nations based on their shared online consumption per month. Thus, for Alexa, this

matrix captured the similarity between nations' overall website traffic ranks; for YouTube and Twitter, it would capture similarities between each pair of countries in ranks of trending videos and trending topics respectively. After applying RBO to the ranked lists from each of the three platforms, each month of data yielded a similarity matrix, the values of which indicated usage similarities between all pairs of countries.

## Network analysis

We treated the similarity matrices as a valued network of countries connected based on similarity in web use and the value of similarity being the strength of the connections. We first descriptively analyzed the network properties, identified each country's centrality score, and identified key hubs and bridges. In this study, the higher the value of a tie the more similar is the said pair of countries. Thus, for centrality we relied on weighted degrees, which lends itself to the most elegant interpretation: A country's weighted degree is the extent to which its online consumption is similar to all other countries. A higher average weighted degree score for a country suggests a higher overall similarity in online consumption between this and all other countries. Beyond focusing on the mean value, we computed the Gini coefficient of each weighted degree distribution to compare the extent to which similarities are unevenly distributed.

We calculated the betweenness centralities of each country for all the six matrices. Since our matrices indicate similarities, we inverted the cell values to convert them into distances (see [39] for details of this algorithm). Nodes with high betweenness act as bridges; many such nodes suggest centralization. In our case, countries with high betweenness are those whose consumption patterns are similar to other dissimilar groups of countries. Conversely, nodes with zero betweenness indicate countries whose strongest and weakest ties to other countries are undifferentiated.

## Quadratic assignment procedure

We used a quadratic assignment procedure (QAP) to evaluate the power of various predictor variables in explaining similarity in web and social media usage. Following the approach by Ng and Taneja [14], we constructed six independent variables which could explain these similarities:

**Language similarity.**　Recent literature [40,41] has used a binary classification to denote whether a pair of countries shares a common native/official language or whether their languages were under the same language family, as proxies for language similarity. However, binary measures are unable to capture linguistic heterogeneity and underestimate the impact of language on web use. Thus, we collected the estimated number of speakers of 460 different languages spoken among the 174 countries, primarily from Ethnologue and cross-checked with other updated sources such as the CIA World Factbook. Ethnologue [42] is a comprehensive catalog of the known spoken languages globally and provides estimated numbers of total language speakers for *all* languages spoken by more than 1% of a country's population (see also [31], for a study using this source).

We calculated the proportion of speakers for each language in each country. For each language, we included people who spoke it as either a first or second language. We quantified language similarity of each pair of countries using cosine similarity:

$$\cos(a, b) = \frac{\sum_{i=1}^{m} a(i)b(i)}{\sqrt{\sum_{i=1}^{m} a(i)}\sqrt{\sum_{l=1}^{m} b(i)}}$$

with $a$ = Country $A$, $b$ = Country $B$, and $m$ = 460 languages. Each country ($a$, $b$ . . .) is a vector with languages {$a_1$, $a_2$, $a_3$. . .}, {$b_1$, $b_2$, $b_3$. . .} as components weighted by proportion of speakers.

**English prevalence.** To account for both the proportion and the actual number of English speakers in each country, we calculated the relative English prevalence between a pair of countries as the multiplication between the multiple of a country's proportion of the English-speaking population and the number of English speakers. We used the formula: $PE_A E_A$ X $PE_B E_B$, where $PE_A$ and $PE_B$ represent the proportion of English speakers in Country $A$ and Country $B$ respectively; $E_A$ and $E_B$ represent the number of English speakers in Country $A$ and Country $B$ respectively.

**Relative size of the internet market.** To account for the effect sizes for both penetration rate and the actual population of Internet users, we calculated, using data from Internet World Stats (data of 2020 Year-Q1 Estimates) [43], the relative size of the Internet market between each pair of countries as the absolute difference between the multiple of each country's Internet penetration rate and Internet population. We used the formula: $PI_A I_A - PI_B I_B$, where $PI_A$ and $PI_B$ represent the Internet penetration rate in Country $A$ and Country $B$ respectively; $I_A$ and $I_B$ represent the number of Internet users in Country $A$ and Country $B$, respectively.

We created a symmetric country-by-country matrix for each of these four variables, where each cell represented the language similarity, relative prevalence of English speakers, the Internet market, and the relative distance respectively between the countries in rows and columns. To compare coefficients, we standardized the values in all matrices to their $z$-scores.

**Shared borders.** Based on the French Research Centre in International Economics' distances and geographical database [44], we created a fourth matrix in which each cell with a value of 1 indicated a shared land border between a pair of countries and 0 indicated otherwise. We did not use physical distances as recent studies of internet use on similar scales (e.g., [14]) have shown the usefulness of shared borders over physical distance as a measure.

**The effect of the United States and China.** We created two additional matrices of countries, in which a value of 1 indicated if one of the countries was the United States in the first matrix, and if it was China in the second matrix. Henceforth, we term these matrices as "US effect" and "China effect." Since YouTube and Twitter are blocked in China, we only modelled the China effect on website traffic.

Finally, for each month we modeled the association between these contextual variables, and the three focal matrices, using regressions based on quadratic assignment procedures (QAP). QAP is a statistical test to determine whether multiple matrices on the same set of nodes are able to explain a "dependent" matrix [45]. The predictor matrices may include matrices derived from combinations of attributes. The regression coefficients are estimated in the same way as for a standard regression (i.e., converting all matrices to vectors based on their cell values). For each of the estimates, however, the null distribution is generated by permuting the rows or columns focusing on the given matrix, while the other matrix remains unpermuted. Thus, unlike in a standard OLS regression, the significance is determined based on a nonparametric distribution derived from a bootstrap-like procedure. The more the number of permutations, the more sensitive the test.

## Results

In all, we analyzed six similarity matrices, two separate months—September 2019 (main study, reported here) and November 2019 (replication data, reported in supplement)—for each of the three platforms. Treating these as networks with countries as nodes and their usage similarities as strengths of links, we computed the weighted degree and the betweenness centralities for each country.

**Table 1. Descriptive statistics for matrices of Alexa, YouTube, and Twitter (September).**

| | With all available countries | | | | | With 59 common countries[*] | | | |
|---|---|---|---|---|---|---|---|---|---|
| **September** | | | | | | | | | |
| *Platforms (# of countries)* | **Weighted degrees (density)** | | | | **Proportion of nonzero betweenness centralities** | | **Weighted degrees** | | |
| | *Mean* | *S.D.* | *Gini* | *# of clusters* | | | *Mean (density)* | *S.D.* | *Gini* |
| Alexa (124) | 39.11 | 6.25 | .08 | 3 | .03 | Alexa | 20.74 | 2.92 | .07 |
| YouTube (98) | 7.19 | 3.22 | .25 | 4 | .55 | YouTube | 4.11 | 2.10 | .29 |
| Twitter (61) | 1.54 | 0.70 | .25 | 5 | .56 | Twitter | 1.45 | 0.66 | .25 |

[*] Common countries among three platforms include Algeria, Argentina, Australia, Austria, Bahrain, Belarus, Belgium, Brazil, Canada, Chile, Colombia, Denmark, Dominican Republic, Ecuador, France, Germany, Ghana, Greece, India, Indonesia, Ireland, Israel, Italy, Japan, Jordan, Kenya, Kuwait, Latvia, Lebanon, Malaysia, Mexico, Netherlands, New Zealand, Nigeria, Norway, Oman, Pakistan, Panama, Peru, Philippines, Poland, Portugal, Puerto Rico, Qatar, Russia, Saudi Arabia, Singapore, South Africa, South Korea, Spain, Sweden, Switzerland, Thailand, Turkey, Ukraine, United Arab Emirates, United Kingdom, United States, and Vietnam (59).

Table 1 reports the summaries of the weighted degrees scores of nations in September 2019. Weighted degrees indicate the overall similarity of a country's usage with all other countries. Based on the average weighted degree, which in a weighted graph is also a measure of density, we observe Website traffic (Alexa) between countries, on average, to be more similar than YouTube trends, which in turn were more similar than Twitter trends. We also computed the Gini coefficient of each weighted degree distribution, which shows that similarity in website traffic, with the lowest Gini value, was the most unevenly distributed of the three. In other words, a few countries were much more similar to the rest in their YouTube usage compared to most other countries, but this inequity was a little less pronounced on Twitter and even less so on the Alexa website traffic.

Fig 1 shows the network betweenness centralities of website traffic (Alexa), YouTube, and Twitter in September. Website traffic similarities had a much lower proportion of countries with non-zero betweenness scores (.03) compared to YouTube (.55) and Twitter (.56). Website traffic patterns are thus more decentralized than social media usage. Thus, despite overall higher similarity in website traffic, YouTube and Twitter consumption was more centered around countries that act as bridges or in other words have highly similar usage trends to most other countries. The high betweenness scores of Singapore, across all three modes of online consumption studied, confirms its status as a multicultural society, which serves as a bridge between the western world and south-east Asia. Likewise, the United Arab Emirates which has a significant immigrant population from South Asia also has high betweenness scores in all three networks. Consistent with the distribution of betweenness scores, website traffic had a lower tendency for clustering, compared to Twitter and YouTube. In Fig 2(a)–2(c), we visualize these graphs with nodes colored by community membership. The community structure, detected using a modularity based Louvain algorithm accords with the overall centrality scores.

To explain these similarities, we built a series of regression models using quadratic assignment procedure (QAP) via the *statnet* family of packages in the R statistics language ([46], 2003–2020). Most variables were only slightly correlated with each other, allaying serious concerns about multicollinearity (see S4 Table). However, some moderately high correlations needed more attention. For instance, Internet market size was moderately correlated with the US effect ($r = .64$) and China effect ($r = .78$). Likewise, English prevalence was moderately correlated ($r = .46$) with the US effect, as the United States had a high overlap in English language

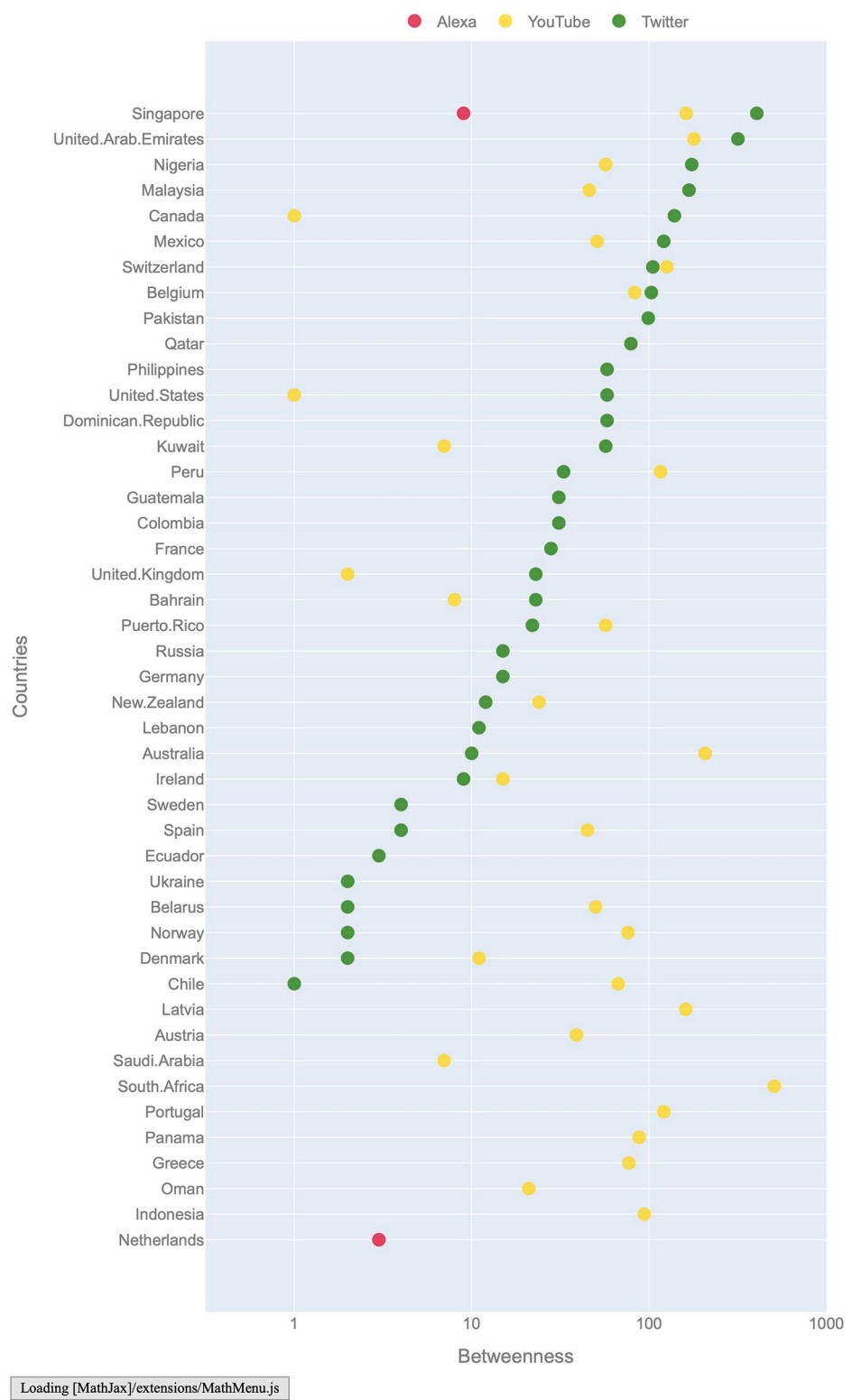

**Fig 1. Network betweenness centralities of websites (Alexa), YouTube, and Twitter in September (Countries with at least one non-zero betweenness, in log scale, sorted by Twitter).**

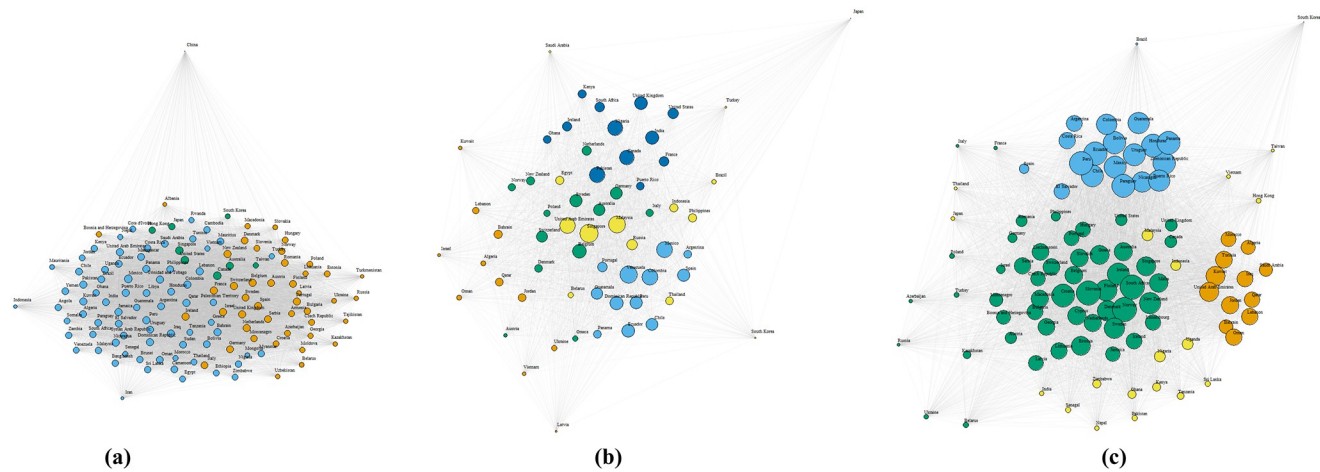

**Fig 2. Visualizations of country similarity networks based on different modalities of online consumption.**

speakers with all other countries. We addressed these correlations by building a series of models in blocks, first including only one of the two variables from these correlated pairs.

Table 2 reports the final blocks of QAP regressions of all three platforms (S5–S7 Tables presents all blocks of QAP regressions). For website traffic, similarity in language composition significantly predicted similarity ($b$ = .04, $p$ < .001), and so did shared borders ($b$ = .02, $p$ = .025), and difference in Internet market size ($b$ = −.02, $p$ = .005). We added the indicator matrices for the US and China effect in the second model. The effect of the United States was not significant (US: $b$ = .04, $p$ = .308), but the effect of China was significant (China: $b$ = −.25, $p$ = .001). We built a third model where we added English prevalence and its effect was insignificant ($b$ = .08, $p$ = .171), nor did it impact the coefficients or the standard errors of the US effect matrix, alleviating potentially concerning effects of multicollinearity. The QAP regression explained 39% of the total variance.

**Table 2. QAP regressions for popular websites (Alexa)/ videos (YouTube)/ topics (Twitter) similarity across countries (Final block, September).**

| Variables | b | | |
|---|---|---|---|
| | **Alexa (N = 124)** | **YouTube (N = 98)** | **Twitter (N = 61)** |
| Intercept | 0.31*** | 0.06** | 0.02** |
| Language composition | .03*** | .07*** | .01*** |
| Sharing border | .02** | .08*** | .01** |
| Internet market size | −.004 | −.02*** | −.002 |
| US effect | −.04 | .02 | −.02 |
| China effect | −.25*** | NA | NA |
| English prevalence | .002 | −.005** | .005*** |
| $R^2$ | .39*** | .50*** | .28*** |
| Adjusted $R^2$ | .39*** | .50*** | .28*** |

Notes: 1,000 permutations for estimating standard errors.

Coefficients presented are standardized coefficients.

* $p$ < .05

** $p$ < .01

*** $p$ < .001.

For YouTube, similarity in language composition ($b = .07$, $p < .001$), shared borders ($b = .08$, $p < .001$), and difference in Internet market size ($b = -.02$, $p < .001$) significantly predicted similarity in YouTube trending videos. The China effect variable was excluded as YouTube was blocked in China. The US effect in the second model was insignificant ($b = .01$, $p = .802$). English prevalence in the third model was significant but with a negative coefficient ($b = -.005$, $p = .009$). The QAP regression explained 50% of the total variance.

For Twitter, similarity in language composition ($b = .01$, $p < .001$) and shared borders ($b = .01$, $p = .012$) significantly predicted similarity in Twitter topics. However, unlike for YouTube, the difference in Internet market size ($b = -.002$, $p = .319$) was not significant. Twitter, similar to YouTube, is also banned in China, so we excluded the China effect variable. The US effect (in the second model) was insignificant ($b = -.002$, $p = .892$). English prevalence was significant with a positive coefficient ($b = .005$, $p < .001$). The QAP regression explained 28% of the total variance.

Comparing three platforms, the variance explained (R-squared) was the highest for YouTube (50%), implying that the factors we investigated exerted more influence upon YouTube similarity than website traffic (39%) and were the least influential on similarity in Twitter trends (28%). Language composition and shared borders significantly predicted similarity in usage for all three platforms. However, language similarity had the greatest influence on YouTube usage followed by website traffic and was the least influential in explaining similarity in Twitter use. Internet market size was a significant predictor only for YouTube use similarity but not for Twitter or Website traffic. Thus, on average, a pair of countries which either had a similar language composition or shared a border would have similar website traffic, YouTube, and Twitter use patterns. In contrast, the negative coefficient of Internet penetration indicated that an increase in the gap of Internet penetration rate between pairs of countries was associated with less similar Internet use patterns.

Since YouTube and Twitter are blocked in China, we modelled the China effect just for the website traffic and it was significant ($b = -.25$, $p < .001$). The US effect was insignificant for all three platforms.

English prevalence was significant for Twitter and YouTube with the effect in different directions. English prevalence had a positive influence on Twitter use similarity, but it was negatively associated with similarity in trending videos on YouTube. Countries with large English-speaking populations thus generated similar trending topics on Twitter but watched a relatively distinct set of YouTube videos. English prevalence did not significantly explain similarity in website traffic.

We repeated our analysis for data collected in November 2019 to strengthen the conceptual validity of our model. Results stayed similar (R-squared values for YouTube: 47%; Alexa: 39%; Twitter: 26%) with the same predictors remaining significant as in September 2019, allaying concerns that these effects could be artifacts of a specific month of data collection.

## Conclusion

Since first becoming popular in the 1990s, the Internet has expanded to virtually every country. Although issues of access persist and a global digital divide remains, one can unequivocally ascertain that in most countries of the world, access to the Internet and the World Wide Web has both broadened and deepened in the past two decades. This is especially the case given the ubiquity of social media and the global popularity of a handful of platforms such as YouTube, Facebook, and Twitter. However, our analysis demonstrates that despite global connectivity and access, internet usage remains rather regional.

Just like previous studies on global web use, our analysis suggests that global web use is quite heterogeneous, whether it is website traffic, or in how people use Twitter or YouTube.

More surprisingly, contrary to expectations derived from prior studies on social media usage [36], we find that despite more potential for content to cross national and regional boundaries, social media usage is even less similar overall than the usage of websites. This difference can be partly attributed to the different level and scale of analysis. People in different countries may be visiting the same websites, but still be accessing different content. By contrast, for social media even though they each visit YouTube and Twitter, the two platforms we studied, people seem to watch different videos and tweet about different topics. A second related explanation is the varying scale of data collection. We captured the popularity ranks of 500 websites per country, but as S1 Table indicates the lists of trending videos and trending topics run much longer, though the number of unique items across the three samples is quite comparable.

Despite being more heterogeneous on average, we also find social media platforms to have higher usage concentration, as evidenced by the Gini coefficient. Thus, while people in different countries on average consume very different content on social media, they do tend to gravitate towards a narrow set of popular trends. The distribution of betweenness centralities of countries in our analysis also accords with this interpretation. Many more countries have significant (non-zero) betweenness scores in their Twitter and YouTube use similarities than website usage, suggesting greater overall convergence driven purely by a narrow selection of popular content. Consistent with this, in our data, in many countries, the same YouTube videos trended near the top for several days in a row. S1 Table reports the total number of videos we examined as well as the number of unique among them.

Overall, these findings complicate our present understanding of global social media usage. They simultaneously explain the global popularity of phenomena such as K-pop [28] and also at the same time support those cultural factors such as language, the best differentiators of web use, are even more salient on social media. Phenomena such as BTS do become global sensations, but at least on platforms we studied they are more likely to remain exceptions.

Language and geography have had a strong influence on what people do online and explain much of the variation in countries' web use [14,47]. We also find that language composition and shared borders significantly predict similarity in usage for all three modes of web use. Yet a high similarity in languages explains similar YouTube usage and website use between countries, the effect is an order of magnitude smaller for Twitter. This accords with the general perception of Twitter being a more elite platform, which over represents the more educated upper-class users. Consistent with this inference, English prevalence had a positive influence on Twitter use similarity, but it was negatively associated with similarity in trending videos on YouTube, suggesting that Twitter users around the world are more likely to be conversant in English than the ordinary web user. YouTube, by contrast, is the most visited site in most countries and thus better represents the average internet user who is more likely driven towards culturally proximate content. The outsized role of English on domains such as Twitter and Wikipedia have previously been documented [31]. Our findings caution against generalizing the role of any factor in influencing global web use based on the study of a single domain or platform could be misleading, no matter how ubiquitous or normatively influential that domain is.

There are several reasons for caution when interpreting findings from this study. First, while Alexa provides a ranked list of popular websites, the platform does not offer raw absolute counts of website traffic. YouTube and Twitter trends do not necessarily imply that trending videos or topics generate the most views/counts. Trading off limitations of using geo-tagged videos or tweets, we consider our approach as a better way to collect geographically specific data. Second, YouTube videos with the same content may be uploaded as different videos or presented as different languages; hashtags with the same meanings or evolve from the same events may be presented in different languages in different countries. We are unable to capture those similarities for now and might have underestimated the similarity of YouTube

and Twitter use. Employing human coders or automated tools to translate from different languages can alleviate this concern.

As the internet has grown and evolved tremendously both in form and content, global attention flows have come to be largely dominated by a handful of social media platforms. These provide a global platform for users to interact, thus potentially homogenizing global web use patterns. We find that such concerns do not map onto people's web usage, not at least just yet. Overall, cultural factors continue to render global web usage into highly regional patterns, both in how people in different countries visit websites or use Twitter and YouTube. Of course, their relative role varies.

## Supporting information

**S1 Table. Scales of data collection.**
(DOCX)

**S2 Table. Network betweenness centralities of the first month for Website traffic (Alexa), YouTube, and Twitter.** (Countries with at least one non-zero betweenness, in log scale, sorted by Twitter).
(DOCX)

**S3 Table. Descriptive statistics for matrices of Alexa, YouTube, and Twitter in September and November.**
(DOCX)

**S4 Table. QAP correlations across variable matrices (based on 59 common countries).**
(DOCX)

**S5 Table.** A. QAP regressions for website traffic (Alexa) similarity across countries (September, N = 124). B. QAP regressions for website traffic (Alexa) similarity across countries (November, N = 124).
(DOCX)

**S6 Table.** A. QAP regressions for YouTube trending video similarity across countries (September, N = 98). B. QAP regressions for YouTube trending video similarity across countries (November, N = 98).
(DOCX)

**S7 Table.** A. QAP regressions for Twitter trending topic similarity across countries (September, N = 61). B. QAP regressions for Twitter trending topic similarity across countries (November, N = 61).
(DOCX)

## Author Contributions

**Conceptualization:** Yee Man Margaret Ng, Harsh Taneja.

**Data curation:** Yee Man Margaret Ng.

**Formal analysis:** Yee Man Margaret Ng, Harsh Taneja.

**Investigation:** Yee Man Margaret Ng.

**Methodology:** Yee Man Margaret Ng, Harsh Taneja.

**Resources:** Yee Man Margaret Ng, Harsh Taneja.

**Supervision:** Yee Man Margaret Ng, Harsh Taneja.

**Visualization:** Yee Man Margaret Ng, Harsh Taneja.

**Writing – original draft:** Yee Man Margaret Ng, Harsh Taneja.

**Writing – review & editing:** Yee Man Margaret Ng, Harsh Taneja.

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
