## [Decision Letter · Decision Letter 0]

26 Sep 2022

PONE-D-22-22223Web use remains highly regional even in the age of global platform monopoliesPLOS ONE

Dear Dr. Ng,

Thank you for submitting your manuscript to PLOS ONE. After careful consideration, we feel that it has merit but does not fully meet PLOS ONE’s publication criteria as it currently stands. Therefore, we invite you to submit a revised version of the manuscript that addresses the points raised during the review process. Both reviewers recognized the relevance and merit of your manuscript. They also raised a number of issues with the methodology that need to be addressed.Please address the issue I raised about using modularity to compare networks as well as all the issues raised by the reviewers such as the comparability and the accuracy of the datasets, the choice of centrality measure, and computation of physical distances. Please submit your revised manuscript by Nov 10 2022 11:59PM. If you will need more time than this to complete your revisions, please reply to this message or contact the journal office at plosone@plos.org. Please include the following items when submitting your revised manuscript:A rebuttal letter that responds to each point raised by the academic editor and reviewer(s). You should upload this letter as a separate file labeled 'Response to Reviewers'.A marked-up copy of your manuscript that highlights changes made to the original version. You should upload this as a separate file labeled 'Revised Manuscript with Track Changes'.An unmarked version of your revised paper without tracked changes. You should upload this as a separate file labeled 'Manuscript'.

We look forward to receiving your revised manuscript.

Kind regards,

Alexandre Bovet, Ph.D.

Academic Editor

PLOS ONE

Journal Requirements:

2. In your Methods section, please include additional information about your dataset and ensure that you have included a statement specifying whether the collection and analysis method complied with the terms and conditions for the source of the data.

Additional Editor Comments:

Please clearly explain how you performed the clustering of the networks.

Also, modularity values should not be used to compare "how modular" different networks are. It can only be used to compare the modularity of different clustering of the same network. Indeed, even a random graph (i.e. a graph without community structure) can have a partition with a modularity close to 1. Moreover modularity values depend on the size of the networks.

Please remove the modularity values from all the tables comparing different networks and the corresponding discussions. You can leave the number of clusters.

Reviewers' comments:

Reviewer's Responses to Questions

**Comments to the Author**

1. Is the manuscript technically sound, and do the data support the conclusions?

Reviewer #1: Partly

Reviewer #2: Partly

2. Has the statistical analysis been performed appropriately and rigorously? 

Reviewer #1: Yes

Reviewer #2: Yes

3. Have the authors made all data underlying the findings in their manuscript fully available?

Reviewer #1: Yes

Reviewer #2: Yes

4. Is the manuscript presented in an intelligible fashion and written in standard English?

Reviewer #1: Yes

Reviewer #2: Yes

5. Review Comments to the Author

Reviewer #1: Web use remains highly regional even in the age of global platform monopolies (PONE-D-22-22223) raises the question, Has the Internet and/or social media really altered the tendency for people to consume content that is from or for their region due to language and geographical barriers? The authors analyzed the similarities among a hundred countries’ web use patterns simultaneously across the 16 most popular websites, and country specific trends from YouTube and Twitter. They report similar to past research that global web use remains highly regional. This is a very interesting manuscript. The authors should be congratulated for their use of data that is more granular than prior research for language similarity, the examination of the 500 most visited website for each country and multiple data sources (websites, Twitter and YouTube viewing). While the research was carried out carefully, there are a number of issues that prevent its publication. The two most significant are missing citations from the literature review and the need for a complete network analysis, including justification of the use of the chosen centrality measure and network graphics. However, with the changes recommended below, I see no reason why it cannot be published in PLOS ONE.

In the literature, I recommend that the authors review the recent research by Barnett and his colleagues, which are relevant to this research. In particular, I find it surprising that they have not cited Barnett & Benefield (2015), which examined international Facebook ties. The references to these articles are listed below.

Barnett, G.A., & Benefield, G., Predicting International Facebook Ties Through Cultural Homophily and Other Factors, New Media & Society, 19(2), 217-239. 2015.

Barnett, G.A., Ruiz, J.B., Xu, W., Park, J.Y., & Park, H.W., The World is Not Flat: Evaluating the Inequality in Global Information Gatekeeping through Website Co-Mentions. Technological Forecasting & Social Change, 117, 38-45. http://dx.doi.org/10.1016/j.techfore.2017.01.011

Barnett, G.A., Ruiz, J.B., Xu, W., Park, J.Y., & Park, H.W., The World is Not Flat: Evaluating the Inequality in Global Information Gatekeeping through Website Co-Mentions. Technological Forecasting & Social Change, 117, 38-45. http://dx.doi.org/10.1016/j.techfore.2017.01.011

Barnett, G.A., & Algara, C., Diachronic Equivalence: An Examination of the International News Network. Social Network Analysis and Mining, 9(6), 2019. DOI: 10.1007/s13278-019-0549-y

In the methods section:

P 10. L115-124. When reading this, I questioned the use of Twitter. It is primarily an English language site. However, the results justify its inclusion. But it should be discussed here that since it is primarily an English language site, that you expect those results.

P13. L199. What measure of centrality did you use? Average weighted degree score? Why did you not use eigenvalue or closeness ? Eigenvalue centrality is typically considered a better measure of overall centrality. Here you should explain betweenness in detail, since you rely on its measure to draw your conclusions.

P14. L250. Perhaps, you should have added the physical distances between capitals or largest. cities. This would account for islands and nearby countries that don't share a border and the variance in distances among countries that don’t share common borders.

In the results section:

P15. What are the densities of the three networks? This would help network scholars understand the results. Also, I would include the network graphics with the countries labelled for all three networks. It would make the results much easier to understand. We could then see the various communities of nations and those that occupy the betweenness positions. In particular, as Barnett & Benefield show, these communities are typically formed by common language and geographical proximity. These would help support your argument.

P15. L286. You haven't introduced the notion of separate communities of nations. You should and then discuss the meaning of modularity. Note that Barnett & Benefield identify communities of nations based on Facebook ties. Further, this would help explain the results displayed in Figure 1. For example, Singapore connects China, Europe, U.S, India and other English-speaking countries. Canada bridges Chinese, French and English-speaking communities. These communities of nations could then be colored differently in the recommended network graphics.

Minor Issues:

P10. L. 108. Missing period

P18. L. 350 twitter should be capitalized

Reviewer #2: The paper provides convincing evidence to the scientific debate on the global patterns of web and global platform use. The research question is withouth doubt timely and important, and the chosen set of data sources and the applied methodologies (in general) are rather unique and innovative. This is a well-written paper with just a few shortcomings. The RBO, regressions and other network metrics are appropriately used.

My questions and suggestions to the authors are the following:

Comparability of the three data sources given the very different geographical coverage.

The number of countries in the Alexa-, YouTube- and Twitter-based data is very different. They represent three distinct clusters of the global internet, but the analysis does not really take this into account and compares the three databases on a one-to-one basis. (E.g. in Table 1. The trends for Alexa, Youtube and Twitter are important, but directly comparing the indicators might be misleading becase they refer to very different geographical regions of the world.) I suggest the authors to run the analysis only on countries that are included in all three data sources and report the results (at least in the supplementary material).

Without this step, I think conclusions such as "it appears that consumption of content on social media platforms could overall be more homogeneous globally than website traffic" could be somewhat misleading. This sentence implies that global website traffic is compared to the same global social media platform use, but this is not true because of the very different country coverage of the two sources.

Some other smaller questions that should be clarified in the text:

How accurate is data from Alexa for the individual countries? Are there any risks that accuracy and valilidity is not the same across countries? (e.g. most of the Alexa toolbar installations were coming from the US, and in several major internet user countries it was much less which might led to certain biases.) This should be briefly touched in the text.

Where is the data for Ethnology coming from? Probably there is no better source for language data, but I did not find any information about the methodology behind this database. (Again, this is a question regarding the accuracy of the datasets.)

Why do the numbers for Youtube differ so much between September and November? Are there any technical or other reasons for this?

6. PLOS authors have the option to publish the peer review history of their article (what does this mean?). If published, this will include your full peer review and any attached files.

Reviewer #1: No

Reviewer #2: No

---

## [Author Response · Author response to Decision Letter 0]

28 Oct 2022

We have attached our responses in a response letter as a separate file submitted along with the manuscript.

---

## [Decision Letter · Decision Letter 1]

21 Nov 2022

Web use remains highly regional even in the age of global platform monopolies

PONE-D-22-22223R1

Dear Dr. Taneja,

We’re pleased to inform you that your manuscript has been judged scientifically suitable for publication and will be formally accepted for publication once it meets all outstanding technical requirements.

Kind regards,

Alexandre Bovet, Ph.D.

Academic Editor

PLOS ONE

Additional Editor Comments (optional):

Reviewers' comments:

Reviewer's Responses to Questions

**Comments to the Author**

1. If the authors have adequately addressed your comments raised in a previous round of review and you feel that this manuscript is now acceptable for publication, you may indicate that here to bypass the “Comments to the Author” section, enter your conflict of interest statement in the “Confidential to Editor” section, and submit your "Accept" recommendation.

Reviewer #1: All comments have been addressed

Reviewer #2: All comments have been addressed

2. Is the manuscript technically sound, and do the data support the conclusions?

Reviewer #1: Yes

Reviewer #2: Yes

3. Has the statistical analysis been performed appropriately and rigorously? 

Reviewer #1: Yes

Reviewer #2: Yes

4. Have the authors made all data underlying the findings in their manuscript fully available?

Reviewer #1: Yes

Reviewer #2: Yes

5. Is the manuscript presented in an intelligible fashion and written in standard English?

Reviewer #1: Yes

Reviewer #2: Yes

6. Review Comments to the Author

Reviewer #1: The authors have addressed all my criticisms and therefore, the manuscript is now worthy of publication.

Reviewer #2: Thank you for the feedback to the reviewer's comments. As the result of of clarifications and additions to the methodology, the text became more straightforward and better understandable. Adding graph visualizations and including information on the common country set of the databases made important improvements to the text. I recommend to accept the manuscript for publication.

7. PLOS authors have the option to publish the peer review history of their article (what does this mean?). If published, this will include your full peer review and any attached files.

Reviewer #1: No

Reviewer #2: No

---

## [Editor Report · Acceptance letter]

20 Dec 2022

PONE-D-22-22223R1 

Web use remains highly regional even in the age of global platform monopolies 

Dear Dr. Taneja:

I'm pleased to inform you that your manuscript has been deemed suitable for publication in PLOS ONE. Congratulations! Your manuscript is now with our production department. 

Kind regards, 

on behalf of

Prof. Alexandre Bovet 

Academic Editor

PLOS ONE